# Allogeneic CAR-T Therapy Technologies: Has the Promise Been Met?

**DOI:** 10.3390/cells13020146

**Published:** 2024-01-12

**Authors:** Caroline Lonez, Eytan Breman

**Affiliations:** Celyad Oncology SA, 1435 Mont-Saint-Guibert, Belgium; ebreman@celyad.com

**Keywords:** allogeneic, chimeric antigen receptor, off-the-shelf, gene editing

## Abstract

This last decade, chimeric antigen receptor (CAR) T-cell therapy has become a real treatment option for patients with B-cell malignancies, while multiple efforts are being made to extend this therapy to other malignancies and broader patient populations. However, several limitations remain, including those associated with the time-consuming and highly personalized manufacturing of autologous CAR-Ts. Technologies to establish “off-the-shelf” allogeneic CAR-Ts with low alloreactivity are currently being developed, with a strong focus on gene-editing technologies. Although these technologies have many advantages, they have also strong limitations, including double-strand breaks in the DNA with multiple associated safety risks as well as the lack of modulation. As an alternative, non-gene-editing technologies provide an interesting approach to support the development of allogeneic CAR-Ts in the future, with possibilities of fine-tuning gene expression and easy development. Here, we will review the different ways allogeneic CAR-Ts can be manufactured and discuss which technologies are currently used. The biggest hurdles for successful therapy of allogeneic CAR-Ts will be summarized, and finally, an overview of the current clinical evidence for allogeneic CAR-Ts in comparison to its autologous counterpart will be given.

## 1. Introduction

These last decades, immunotherapy has become an important treatment option for patients with cancer indications [1,2]. Among the most promising options, T-cells engineered to express chimeric antigen receptors (CAR) aim to strengthen the power of T-cells to recognize and eliminate tumor cells in a human leukocyte antigen (HLA)-independent manner. Since 2017, six CAR-T products have been approved by the US Food and Drug Administration (FDA) in the United States and other countries, and two CAR-T products are approved in China by the National Medical Products Administration [3]. All products are aimed for patients with advanced or resistant large B-cell lymphoma, acute lymphoblastic leukemia or multiple myeloma, where outstanding results were obtained with overall response rates reaching up to 100% objective response rates in some cases [4,5]. Nevertheless, challenges remain for these cell therapies, including the low durability of responses, severe adverse events, low effectiveness in the context of solid tumors, and limitations due to manufacturing of a highly personalized product [4,5,6,7].

CAR-T therapies in advanced stage of development, including those currently marketed, are of autologous origin, whereby peripheral blood cells are taken from the individual receiving treatment to be engineered into CAR-Ts before being reinfused to the patient. The variability among patients in the initial material, due to the patient’s prior treatment and disease history, may result in disparities in efficiency or yield of the end product, leading to a 2–10% manufacturing failure rate [8] and in treatment deprivation for a patient who has already undergone the apheresis process. Another obstacle arises from the logistics, planning and increased expenditures associated with tailored medicines, which necessitate creating and releasing a single product for each patient. The manufacturing, testing and release process itself is time-consuming, and the logistical challenge in shipping cells back and forth between the treatment site and cell production facilities—which usually follows a centralized manufacturing model—poses a significant concern for individuals with rapidly progressive or aggressive cancers. The development of allogeneic and/or ‘off-the-shelf’ CAR-Ts from healthy donors allows many of these limitations to be overcome by contributing to scalability and direct access to CAR-T therapies, providing a readily available therapeutic solution for multiple patients (Figure 1).

Whilst allogeneic therapies are attractive new treatment opportunities, their main downside is the risk of a potential life-threatening toxicity called “graft-versus-host disease” (GvHD) that is triggered by recognition of the patient’s healthy tissues by the T-cell receptor (TCR) present on the surface of allogeneic CAR-Ts. To minimize this risk, selection of T-cell sources presenting low TCR signaling capacity can be considered (see Section 3). Most often, the manufacturing process of allogeneic CAR-T therapies include an engineering step that aims to eliminate or blunt the signaling or the expression of the TCR using specific technology (see Section 4). As a result, the engineered allogeneic CAR-Ts fail to recognize the patient’s healthy tissue as foreign, thereby preventing GvHD.

Another challenge to overcome is the opposite scenario, where the patient’s immune system swiftly rejects any transferred allogeneic cell, called the host-versus-graft (HvG) reaction, thereby limiting the persistence of allogeneic CAR-Ts. For this, too, further engineering of CAR-Ts is needed.

Here, we review potential sources of allogeneic cells for CAR-Ts and focus on advantages or inconveniences of using existing technologies to establish “off-the-shelf” allogeneic CAR-Ts with low alloreactivity, including the most studied and developed gene-editing technologies, but also other non-gene-editing technology alternatives.

## 2. Source of Allogeneic Cells

The potential of allogeneic CAR-T lies largely in the ability to mass-produce CAR-Ts that are as efficient and potent as their autologous counterpart. One of the crucial factors in the manufacturing of allogeneic CAR-Ts lies in the source material used for the final product. 

Currently, the most frequently used allogeneic cell source for CAR-T manufacturing involves using peripheral blood mononuclear cells (PBMCs) from a random healthy donor. More rarely, other cell sources are used such as umbilical cord blood (UCB) or a renewable cell source such as induced pluripotent stem cells (iPSCs).

### 2.1. PBMCs

The most frequent source for the manufacturing of allogeneic CAR-Ts is PBMCs collected from healthy donors, where T-cells are isolated and expanded. This allows for the creation of multiple vials from a single apheresis product that can be easily used in a very rapid and standardized manufacturing protocol [9,10]. Another manner by which CAR-T can be manufactured is via isolation of stem cells from PBMCs [11], which can be further activated and transduced into CAR T-cells. The advantage of using hematopoietic progenitor cells is their ability to self-renew; however, as their absolute numbers are limited, other strategies may be needed for their enrichment, such as CD34+ mobilization, similarly to what is performed for autologous stem cell transplantation [12]. This also allows for the generation of a bank of cells that express different human leukocyte antigen (HLA) subtypes to potentially match the donor HLA to that of the patient [13]. The selection of donors on the basis of their immune characteristics is likely to be a key factor in decreasing the heterogeneity in the final manufactured product and lower the risk of GvHD.

### 2.2. UCB

The use of UCB was shown to be associated with reduced incidence and severity of GvHD, making it a potentially more tolerable source material than PBMCs for allogeneic T-cells and allowing for less stringent HLA matching [14]. Furthermore, UCB is an enriched source of hematopoietic stem cells (HSCs), which are able to self-renew and can be used to differentiate into T-cells, although there is a limit to their total number [15,16].

Interestingly, T-cells isolated from UCB have a unique antigen-naive status which is probably linked to the decreased alloreactivity observed in UCB grafts [17,18]. Furthermore, UCB T-cells are characterized by impaired nuclear factor of activated T-cells (NFAT) signaling and reduced activity, which most likely further contributes to the reduced GvHD [19].

However, an obvious drawback of UCB is its limited availability compared to other cell sources.

### 2.3. Induced Pluripotent Stem Cells (iPSCs)

T-cells derived from iPSCs can also be used as a source of CAR-Ts [20]. In theory, iPSCs have an unlimited capacity for self-renewal, thus allowing them to be banked and used indefinitely [21]. A bank of iPSC lines with different homozygous HLA combinations could be generated to minimize the risk of allorejection of CAR-T derived from iPSCs [22]. An advantage of using iPSCs is that CAR T-cells can be generated from a single iPSC clone with the capacity for clonal expansion, and therefore, the genetic modifications they undergo would be homogeneous in the final cell population [23]. However, the quality controls should be strict because undifferentiated proliferating iPSCs may compromise product safety, since they could induce important adverse effects such as teratomas [24].

iPSCs can be developed from different cell types, such as fibroblasts or lymphocytes, that are reprogrammed into a less differentiated cell by inducing the expression of specific factors. For example, Iriguchi et al. generated iPSCs from an antigen-specific cytotoxic T-cell clone, or from TCR-transduced iPSCs, as starting material [25]. These iPSCs can then in turn be differentiated into T-cells through the addition of several differentiation drivers and/or inhibitors (SDF1α and p58 inhibitors in the above case, for example) to enhance T-cell commitment. While the potential to create a large cell bank that covers a study cohort is appealing, the arduous tasks of T-cell differentiation and selection leading up to the commitment of a single positive T-cell is much more complex then the use of T-cells isolated from either PBMCs or UCB. However, while PBMCs and UCB both offer a heterogenous T-cell population of cells, iPSCs are clonal and thus give rise to a homogenous T-cell population both with the advantages/disadvantages of each. 

The generation of allogeneic CAR-T irrelevant of the starting material faces two major hurdles. The first is the induction of GvHD and the second is the HvG response. Each T-cell expresses a T-cell receptor (TCR), where the majority of T-cells express a TCR composed of an alpha and a beta protein chain that can recognize HLA-peptide complexes on target cells through the direct pathway of allorecognition, thus leading to GvHD [26,27] independent of the CAR.

## 3. How to Prevent Alloreactivity in CAR-Ts by Selecting the Right Cell Population?

The use of allogeneic donor T-cells (CAR or not) that still express a functioning TCR may play a role in anti-tumor effects. This has been clearly demonstrated in leukemia in a process termed graft-versus-leukemia (GvL): after allogeneic stem cell transplantation (SCT), protection from relapse is partly due to donor T-cells that recognize leukemia-specific minor antigens [28]. This may be similar to CAR-T, although the recognition of allo-antigens will likely induce GvHD, as studies assessing both acute and chronic GvHD have clearly established a central role for αβTCR in GvHD pathogenesis [29,30,31,32]. The application of SCT, for example, was not appreciated until T-cell-depleted grafts were assessed to eliminate GvHD [33,34]. These successfully decreased the occurrence of GvHD to extremely low frequencies, although the risk of opportunistic infections and relapse increased substantially [34,35]. While the role of αβTCR in GvHD development is not in doubt, the possible risks and/or benefits in the case of CAR-T therapy are not completely clear, and the development of GvHD may be relatively low [36].

To avoid GvHD, two main approaches exist depending on (i) T-cells that have low or non-reactive TCRs (discussed in this section) or (ii) engineering methods to avoid allorecognition (Section 4). The αβTCR repertoire is selected in the thymus and is educated based on the ability to be tolerant to self-HLA complexes. This tolerance means that the TCR recognizes the self-HLA and responds to non-self peptide. However, in the case of allorecognition, the TCR recognizes both structurally similar HLA-peptide complexes and dissimilar HLA-peptide complexes, therefore allowing for the high frequency of alloreactive T-cells (1 in 10^3^) [37]. It is these alloreactive αβTCRs expressed on T-cells that drive GvHD.

The HLA locus is the most polymorphic region in the human genome, thus leading to many HLA variants in each individual. There are six HLA-class-I molecules and six HLA-class-II molecules, making the matching between donor and patient a complex issue, and although decades of data from transplantation centers have shown that the most important HLAs to match are the class I HLAs A,B and class II HLA-DR [38,39], this still requires a vast bank of cells in order to produce the CAR-Ts, which renders the allogeneic manufacturability rather complicated.

### 3.1. Infusion of Allogeneic CAR-Ts Post or Prior to an Allogeneic Transplantation

Patients treated with allogeneic SCT can be subsequently treated with CAR-Ts generated from the same donor if they relapse. This was performed in a study by Brudno et al., where 20 patients with B-cell malignancies received CD19 CAR-Ts generated from the same donor as SCT with no chemotherapy administered before T-cell infusion. Six patients achieved complete remission and two patients achieved a partial response. No GvHD was reported [40]. These results confirmed previous observations made by other groups [41,42]. In a more recent study, eight r/r B-ALL patients received either HLA-matched (n = 4) or HLA-haploidentical (n = 4) CD19 CAR-Ts immediately preceding an intended SCT [43]. The haploidentical CAR-Ts induced transient or no reduction in peripheral blood leukemia cells with no significant CAR-T expansion, which suggests rejection. In contrast, patients treated with the HLA-matched CAR-Ts exhibited higher complete response rates, although more severe toxic side effects, with no GvHD observed in either group. However, only three out of eight patients reached complete response and only two of the eight patients proceeded to transplant, indicating that while HLA-matched and HLA-haploidentical allogeneic CD19 CAR-Ts are feasible in r/r B-ALL before SCT, other factors besides GvHD need to be considered in clinical applications of allogeneic CAR T-cell infusions.

### 3.2. Memory T-Cells

T-cells with a specific memory phenotype are considered to have a TCR specificity directed to previously detected antigens, which are expected to be different from those of the patient receiving the CAR-T therapy. Interestingly, studies have shown that memory T-cells do not induce GvHD [44]. It is unclear why this is the case, but one possibility is the diversity of the TCR, which is limited in memory T-cells, thus reducing GvHD. One manner by which the T-cell memory and TCR specificity can be further specified is through selection or the development of virus-specific T-cells (VST), as has been achieved in Epstein–Barr virus (EBV)-associated malignancies. Adoptive transfer of HLA partially matched EBV-specific T-cells from healthy donors has had positive results in post-transplant lymphoproliferative disease for example, with response rates of 60–70% and low incidences of toxicity or GVHD [45]. Infusion of EBV-specific T-cells has also been used in patients with Hodgkin’s lymphoma with good tolerance and remission rates [46,47]. The use of viral antigens can enhance the proliferative capacity of the allogeneic CAR-Ts, making them persist longer and possibly enhance their efficacy. This has been shown with cytomegalovirus (CMV)-specific CD19-CAR-Ts that had enhanced in vivo anti-tumor activity following the administration of anti-CMV vaccination [48].

However, all these methodologies require partial matching and thus require the creation of multiple cellular banks. Next to the above-mentioned options, sub-populations of T-cells can be used for the generation of allogeneic CAR-Ts.

### 3.3. T-Cell Sub-Populations

T-cell sub-populations comprise a relatively low percent of the circulating total T-cells (making up anywhere between 0.01 and 10% of T-cells). These sub-populations include double-negative T—cells (DNTs); invariant Natural Killer T-cells (iNKT); cytokine-induced killer (CIK) cells; mucosal-associated invariant T (MAIT)-cells; and lastly, γδT-cells.

#### 3.3.1. Double-Negative T-Cells (DNTs)

DNTs are a rare subset of immune cells that express CD3 but not CD4, CD8, and CD1d-αGalCer [49,50,51]. DNTs comprise about 1 to 5% of human PBMCs and can be isolated and expanded ex vivo under clinically compliant conditions from the peripheral blood of healthy donors [52,53]. Expanded DNTs can express either γδTCR or αβTCR, where the frequency of TCR expressing DNTs can range between 60 and 90% depending on the donor origin.

In a recent study conducted by Vasic et al. the feasibility, safety, and efficacy of DNTs for the development of allogeneic CD19-CAR-T was assessed. The resulting allogeneic CD19-CAR DNTs had the properties of an off-the-shelf cellular therapy and were effective against CD19-expressing hematological and solid malignancies [54]. Pre-clinical studies have thus confirmed the feasibility of DNTs, but whether DNTs will actually yield good results clinically remains to be seen.

A phase I/IIa clinical trial using third-party-donor-derived, genetically non-modified DNTs to treat patients with relapsed/refractory acute myeloid leukemia (AML) showed that the therapy was safe and had a positive efficacy profile [55]. One major concern is regarding the cellular efficacy. Interestingly, Kang et al. have shown that one manner by which the cellular efficacy and persistence of DNTs CARs can be enhanced is through inhibition of the PI3K pathway during manufacturing, Which is something that we and others have seen in αβT-cells as well [56,57].

#### 3.3.2. Invariant Natural Killer T-Cells (iNKTs)

Invariant NKT-cells (iNKTs) are a subset of T-cells that share morphological and functional characteristics of both NK and T-cells. They have a restricted TCR that has a constant α-chain paired with a low-diverse β-chain. iNKTs comprise between 0.01 and 1% of the peripheral blood T-cell population and have shown not to cause GvHD in xenograft models [58,59,60]. They are restricted by CD1d, a glycolipid-presenting HLA-I-like molecule expressed on B-cells, antigen-presenting cells and some epithelial cells [61,62]. The fact that iNKT-cells recognize B-cell lymphomas through CD1d makes them of particular interest for B-cell malignancies [63].

#### 3.3.3. Cytokine-Induced Killer (CIK)-Cells

CIK-cells are a heterogenous population of polyclonal effector T-cells that have functional NK-cell properties. They comprise between 0.01 and 1% of the peripheral blood T-cell population and can be expanded from PBMCs, bone marrow and UCB through a manufacturing process that involves the addition of cytokines like IFN-γ and IL-2 and TCR-activating antibodies [64,65]. CIK-cells have the advantage of exerting non-HLA-restricted cytotoxicity and very low alloreactivity across HLA barriers in comparison with conventional donor lymphocyte infusion [66,67,68]. This was further confirmed by preclinical and phase I/II studies, where the infusion of bulk CIK-cells population was well-tolerated [69,70,71]. In addition to the alloreactivity, the dual activity (of both NK cell receptors and TCRs) gives CIKs an added ability to mediate cytotoxicity and prevent infection, which is a major concern after CAR-T therapy. In a recent clinal trial where relapsed B-acute lymphoblastic leukemia (B-ALL) patients were treated with CD19 CAR CIK-cells, no GvHD was observed, and the cells could be detected up to 10 months after infusion [72]. The overall response rate was 61.5% (13 patients), which is in line with its autologous counterpart.

#### 3.3.4. Mucosal-Associated Invariant T (MAIT)-Cells

MAIT-cells are primarily localized to mucosa-rich regions, comprising a fraction of T-cells distributed throughout the pulmonary (5%), hepatic (20–40%) and intestinal (1–2%) lamina propria, as well as peripheral circulation (1–10%; [73,74,75]). MAIT-cells have a heavily restricted TCR repertoire that consists of TCR alpha variable (TRAV)1 combined with three kinds of TCRA junctionals (TRAJ; TRAJ33, TRAJ12, TRAJ20) and a limited repertoire of β chains in humans [76]. The MAIT TCR can recognize modified derivatives from the vitamin B2 synthesis pathway presented by MHC class I-related molecule MR1 on APCs. MR1 is a conserved molecule, thus making MAIT-cells incapable of inducing strong GvHD in vivo [77]. This has further been shown in clinical studies where MAIT-cells were positively correlated with improved survival and fewer allogeneic adverse events [78].

The use of MAIT-cells for CAR-T has been assessed in multiple pre-clinical studies, and while their efficacy against tumor antigens was significant (as assessed with a mesothelin and a CD19-targeting CAR), significant concerns were raised based on both cellular persistence and manufacturing due to the limited cell number [79,80]. These concerns imply that the use of MAIT-cells clinically may be limited.

#### 3.3.5. γδT-Cells

One other subset of T-cells that is currently being used extensively in both preclinical and clinical studies are γδT-cells (reviewed elsewhere [81,82,83]), which represent 1–10% of circulating T-cells (although they are also prevalent in some epithelial tissues; [84]). The γδT-cells have a unique TCR composed of variable gamma and delta chains and recognize antigens independent of the HLA, leading to low or no risk of GvHD [85,86]. It is this advantage that has made them a popular starting material for the creation of allogeneic CAR-Ts, and at least a dozen trials are currently underway to assess this as a viable option [81,83,87].

Several studies have shown the safety and some efficacy of γδT-cells’ transfusion into cancer patients, thereby relying on the HLA-independent function of γδT-cells (mediated by NKG2D, for example, among others; [88,89]). These studies imply that the use of γδT-cells may prove beneficial as a CAR-T therapy. This observation has led to multiple CAR-T- and TCR-based strategies being employed by companies to improve the efficacy of γδT-cells for cancer immunotherapy. However, the tumor toxicity has been limited, and consistent problems with both persistence and homing in vivo has limited the translation of γδCAR-Ts.

## 4. ‘Off-the-Shelf’ Allogeneic CAR-Ts

### 4.1. Methods to Engineer ‘Off-the-Shelf’ Allogeneic CAR-Ts

The strategies to reduce GvHD by using partially matched allogeneic material, and/or T-cells that have low or no TCR, naturally offer good alternatives, and many CAR-Ts have shown the alloreactivity to be limited or manageable. However, in most instances, allogeneic cells are persistent for a very short amount of time, meaning that the lack of GvHD may be due in part to the lack of persistence. This lack of persistence is driven by multiple-factors, including (i) a resurgence of the host immune response (in most instances, the patients undergo lymphodepletion prior to CAR-T therapy) that in turn rejects the allogeneic cells, (ii) the immune-suppressive tumor microenvironment that may inhibit T-cell proliferation, as well as other factors [90]. This requires additional engineering to circumvent the host immune response and/or the tumor microenvironment. The different methods can be divided into gene-editing technologies and non-gene-editing technologies.

#### 4.1.1. Gene-Editing Technology

The two biggest hurdles in the use of allogeneic T-cells are GvHD and HvG. The former can be avoided by eliminating the TCR, usually through the knockout (KO) of the constant domain of one of its chains (α and/or β), or by replacing some TCR subunits, which impedes its antigen recognition function [91]. However, although this takes care of the alloreactivity, the cells would still be susceptible to HvG. The most common antigens driving HvG are the mismatched donor-HLA-I molecules on the donor cells. These are recognized by the patient αβT-cells that are CD8+ through the direct pathway of allorecognition. By knocking-out the common subunit β2-microglobulin (encoded by the B2M gene), the HLA-I molecule will not be expressed on the cell surface, thus making the cell susceptible to NK-cell lysis [92]. To avoid recognition by NK-cells, different strategies have been developed, most commonly utilizing overexpression of a non-classical HLA-I such as HLA-E or G fusion protein to avoid lysis [93,94].

Other strategies to avoid HvG include (i) CD47 overexpression [95] and (ii) CD52 KO [96]. CD47 is found on both healthy and malignant cells and regulates macrophage-mediated phagocytosis by sending a “don’t eat me” signal to the signal regulatory protein alpha receptor. Upon depletion of HLA-I on CAR-Ts, recognition by both macrophages and NK-cells is triggered. In a recent study by Hu et al., the overexpression of CD47 in allogeneic CD19-CAR-T negated the recognition of NK and macrophages to the absence of HLA on the cell surface, thus avoiding rejection [97]. This approach is currently under investigation in a phase I clinical trial (NCT05878184).

CD52 is protein expressed on the cell surface of many immune cells such as mature lymphocytes, NK-cells, monocytes/macrophages and others [96,98]. The humanized anti-CD52 monoclonal antibody (mAb), alemtuzumab, has been widely used in clinics for the treatment of transplant patients and B-cell chronic lymphocytic leukemia [99,100,101]. Alemtuzumab targets CD52+ T-cells and is capable of both complement-dependent cytotoxicity and antibody-dependent cell-mediated cytotoxicity [100]. Therefore, CD52 KO in allogeneic CAR-Ts can be combined with Alemtuzumab to enhance CAR persistence. However, this will necessitate multiple infusions and close monitoring of the immune system of each patient. This approach has been assessed in multiple clinical trials involving allogeneic CAR-Ts, most notably by Allogene, who have used this in combination with CD70 [102] and CD19 CAR-Ts [103].

Next to recipient CD8+ T-cells that recognize the HLA-I molecules, CD4+ T-cells recognize HLA-II molecules that are expressed on multiple cell types, including activated T-cells [104]. Therefore, once donor CAR-Ts recognize their antigen, they will upregulate the HLA-II expression and become targets for recognition by recipient CD4+ alloreactive T-cells [105]. It is therefore likely that for a persistent CAR-T, the removal of HLA-II becomes necessary. One strategy that can achieve this is through the removal of the CIITA gene, an HLA class II transactivator that controls HLA-II expression [106].

However, it is likely that for the success of allogeneic CAR-Ts, other modifications become necessary to tackle the tumor microenvironment, for example. Different strategies exist to introduce double-stranded DNA breaks (DSBs) that allow for the editing of proteins. These breaks are subsequently repaired in error-prone pathways that can result in insertions/deletions that can disrupt open reading frames. An overview is shown in Table 1.

**Zinc finger nucleases (ZFNs)**—A ZFN is an artificial endonuclease that has a zinc finger protein (ZFP) fused to the cleavage domain of the FokI restriction enzyme [107]. A ZFN is targeted to cleave a chosen genomic sequence. The FokI cleavage domain needs to be dimerized to cut DNA, and because the dimer interface is weak, a construct of two sets of fingers directed to neighboring sequences is needed. The cleavage-induced event caused by ZFN leads to a cellular repair process that mediates the efficient modification of the targeted locus. If the event is resolved via non-homologous end joining (NHEJ), it can result in small deletions or insertions, effectively leading to gene KO. If the break is resolved via a homology-directed repair (HDR), small changes or entire transgenes can be transferred into the chromosome. Because each zinc finger unit recognizes three nucleotides, three to six zinc finger units are needed to generate a specific DNA-binding domain.

The use of ZFNs has multiple challenges such as the specificity of ZFN binding, where some fingers bind equally well to triplets other than their supposed preference. Thus, off-targets can occur, and it is therefore necessary to extensively test ZFNs employed in clinical trials [108,109]. Furthermore, the efficient delivery of ZFNs and donor DNA will naturally be different among applications, and biological variations in the availability of particular DNA repair pathways may affect the outcome.

Current clinical trials involving ZFNs include the knockout of the CCR5 gene, which is the coreceptor for HIV-1 (e.g., NCT02388594, NCT00842634, NCT01044654, NCT01252641, or NCT02225665) [110]. ZFNs are also currently being used for the targeting of the glucocorticoid receptor in IL13Rα2-targeting CAR-T in an allogeneic setting, where infusion of the CAR has led to dexamethasone-resistant effector activity in six patients with unresectable recurrent glioblastoma [111].

**TALEN**—TALENs are similar to ZFNs in that they are heterodimeric nucleases that contain a fusion between the FokI restriction enzyme and a transcription activator-like effector (TALE) DNA-binding domain. The amino acid repeat variable di-residues (RVD) are two hypervariable amino acids that make up part of the sequence that mediates the binding of TALE to DNA [112]. This greatly simplifies TALEN design. The TALENs’ monomeric architectures are developed by fusing TALE domains to a sequence-specific catalytic domain derived from the homing endonuclease (HE) I-TevI, resulting in a Tev-TALE monomeric nuclease [113]. 

Currently, multiple CAR-Ts have been developed using TALEN for the purpose of creating allogeneic CAR-Ts. TALEN has been used to knockout both TRAC and CD52 in UCART19 (a CD19-targeting CAR-T), as assessed by Allogene Therapeutics. Similarly, Cellectis has assessed multiple CAR-Ts such as CAR-Ts targeting CD123 [114], CD22 [115] and CS1 [116]. In all candidates, TRAC was disrupted, but multiple strategies were shown to enhance cellular persistence. Among those, CD52 and B2M have been discussed previously. However, an additional target is CS1 (SLAMF7), which in this instance is specifically removed to inhibit fratricide by the CAR-Ts.

**MegaTALs**—MegaTALs are a short TALE domain that fused to the homing endonuclease (HE). The artificial chimeric nucleases derived from HEs can be engineered to target specific sequences within the genome [117,118,119]. This fusion increases the specificity and activity of the MegaTALs [120]. Currently, to our knowledge, no clinical trials are utilizing MegaTALs for allogeneic CAR-Ts.

**Clustered regularly interspaced short palindromic repeats (CRISPR)**—The CRISPR system is derived from microbial adaptive immune system. It combines a nuclease and a short RNA. The specificity of the CRISPR system is not through the protein-DNA interaction (like the above) but rather RNA-DNA base pairing. A 20 nucleotide RNA that is complementary to the target DNA(termed single guide RNA; sgRNA) is responsible for the specificity. However, due to the system, off-targets are tolerated [121,122]. The most common nuclease is Cas9 [123]. CRISPR/Cas9 is the most widely used because it has demonstrated a remarkably low rate of off-target mutagenesis in T-cells [124,125]. In addition, a specific high-fidelity Cas9 mutant, called eSpCas9, did not cause any detectable off-target effect, making it an even safer technology [126,127].

CRISPR/Cas9 has been used to KO multiple targets to inhibit both GvHD and HvG, focusing on TRAC, B2M, CD52 (as previously mentioned); however, multiple preclinical studies have also shown that the KO of many other genes can play a role in cellular persistence and efficacy, thus giving rise to the need for multiplexing (as reviewed by [128]). However, although CRISPR/Cas has become the standard technology for use in adoptive cell therapy, it is important to note its risks and limitations (Table 1). The first major risk is the issue of off-targets. Well-designed sgRNAs are intended to induce gene insertions or deletions, disrupting protein-coding sequences and establishing functional gene KOs, but they can also induce off-target mutations at regions similar to the target sequence. These off-target mutations have the potential to disrupt the normal genomic sequence, leading to compromised cellular functions and a high risk of cell death [129]. The second risk occurs after on-target CRISPR/Cas KO and relates to unwanted mutations that can occur following DSB-induced repair [130]. The DSBs can lead to chromosomal abnormalities such as chromosomal loss [131], and/or chromosomal translocations/rearrangements [132]. These risks are further increased when multiplexing several genes with Cas9 nuclease; therefore, a lot of work has gone into multiplexing with an effort to reduce this risk. Through the use of a catalytically dead Cas9 or base editor technology, modifications can be made to optimize and improve the limitations of CRISPR/Cas9 [133].

Although multiplexing these in unison becomes increasingly difficult, the relative improvements seen, when such targets are removed, does imply that allogeneic CAR-Ts may need more engineering to become long-lasting CAR-Ts.

**CRISPR/other Cas**—The most widely used CRISPR-Cas system is CRISPR/Cas9; however, there are multiple systems, which are generally divided into two classes (class 1 and 2), and subsequently subdivided into six types (types I through VI). Class 1 (types I, III and IV) systems use multiple Cas proteins, while class 2 systems (types II, V and VI) use a single Cas protein [134]. The class 1 CRISPR/Cas systems comprise 90% of all identified CRISPR/Cas loci. Class 2 comprises the remaining 10% and is almost exclusively in bacteria [135]. Cas9 (type II) still presents challenges, mostly due to the possibility off-targets and difficulty in delivering ribonucleoprotein particles [134]. The second most utilized Cas is Cas12a (type V). It has substantial differences in comparison with Cas9 in multiple aspects, one of which is a higher gene repression in the template strand of the target DNA than SpdCas9 [136]. It may also be easier to multiplex in comparison with Cas9 [137]. However, both Cas 9 and 12a suffer from a dependence on host cell DNA repair machinery, meaning the induction of DSB and induction of repair. Although both technologies have been used successfully to insert specific DNA into the genomic loci, their efficiency differs between cell types [138,139,140]. Furthermore, DNA repair through HDR is also related to active cell division, meaning that cells that do not divide (like neurons) render the tools ineffective.

One such example is Cas-Clover, which relies on an RNA-guided endonuclease (termed Cas-Clover) that consists of a Clo051 nuclease domain that is fused with a catalytically dead Cas9. These changes lead to fewer off-targets compared to CRISPR/Cas9 and a better safety profile (genomic stability and normal karyotype; [141]). However, naturally, other possibilities exist, such as the use of meganucleases that can be mutated individually in order to change the specificity but without disrupting the catalytic efficiency [142]. This approach is currently being assessed with the ARCUS platform in a phase I clinical trial that will be discussed below.

Recently, CRISPR-Cas12a was successfully used in combination with CRISPR-Cas9 to generate simultaneous genetic manipulations for the generation of allogeneic CAR-Ts. Combining both Cas12a and Cas9 led to triple-edited CAR-Ts that resulted in TCR- and HLA-I/II-negative CAR-Ts resistant to allogeneic stimuli [143]. However, due to the nature of DSBs explained above, and the high safety concern when multiplexing CRISPR/Cas, a secondary methodology was necessary to achieve a safe CAR-T and minimize DSBs. This technology is base-pair editing.

**Base-pair editing**—Base editing involves the use of CRISPR-Cas9 (or other Cas) together with avoidance of DNA DSBs during genetic modification. Fusing a single-strand DNA (ssDNA) deaminase enzyme to a catalytically inactive Cas9 variant leads to there being only an ssDNA cut (nick). The Cas9-mediated nicking of the genomic DNA means that a short stretch of ssDNA is exposed to the attached deaminase that can convert the selected bases within their target window [144]. Many improvements have been conducted since the first report on cytosine base editors (BE), and these have yielded novel base editors that reduce unwanted byproducts, improve the targeting scope and allow the editing of different bases [145]. Currently, four possible transition mutations can be installed: C→T, A→G, T→C and G→A.

The added safety and possibility to multiplex gene KO through CRISPRs makes this approach very interesting for CAR-Ts. A proof of concept for the approach was shown by Diorio C et al. using an allogeneic CD7 CAR-T for T-cell acute lymphoblastic leukemia (T-ALL). Here, base editing was used in combination with CRISPR-Cas9 to target four genes, namely CD52 (to enable lymphodepletion with alemtuzumab); TRAC (removal of the TCRα chain, GvHD); CD7 (to inhibit fratricide); and PDCD1 (PD1-receptor—an immune-checkpoint inhibitor) successfully [146], currently under clinical evaluation (NCT05885464). Importantly, the CD7 CAR-Ts functioned well and showed no detectable translocations or karyotypic abnormalities. Similar base-pair-edited CD7 CAR-Ts were assessed in a phase I clinical trial. Preliminary results reported one patient to be in leukemic remission, one that received SCT while in remission and the third developed an opportunistic fatal fungal infection. Other adverse events included cytokine release syndrome and multilineage cytopenia [147].

Multiple publications have shown that BEs can induce transcriptome-wide off-target RNA editing [148,149] and genome-wide off-target DNA editing [150,151], as well as unexpected nucleotide conversions [152]. However, whether these risks are applicable to patients remains to be seen [153].

#### 4.1.2. Non-Gene Editing

The biggest concern with gene editing is the complexity involved in removing multiple genes (multiplexing) while keeping the safety concerns to a minimum. We developed two non-gene-edited approaches: (i) The first is based on a TCR inhibitory molecule (TIM) that, upon incorporation with the T-cell DNA, competes with TCR elements rendering the TCR unresponsive [10]. This approach was used together with an NKG2D-based CAR and assessed in metastatic colorectal cancer [154]. (ii) The second uses an miRNA scaffold targeting CD3ζ, which has led to a complete abolishment of TCR from the cells [155]. This approach was assessed in a phase I clinical trial using a BCMA-targeting CAR-T in a relapse/refractory multiple myeloma patient cohort. 

Another approach includes intracellular retention of TCR/HLA-I to prevent GvHD/HvG. There are multiple methods to retain components in the endoplasmic reticulum (ER), including using a peptide (such as KDEL) that is associated with the ER retention domain. Then, by combining said peptide with an scFv targeting the TCR, for example, all TCRs will be retained in the ER [156].

While the argument for the removal of the TCR is clear, it is unclear which factors govern cellular persistence in an allogeneic setting. While the usual suspects (HLA-I/II) naturally play a role, other proteins are possibly involved in HvG. Furthermore, other cellular processes such as metabolic regulation may affect cellular persistence in an allogeneic setting. Current results suggest that additional modifications are needed to achieve success in an allogeneic setting. In this regard, the ability to multiplex multiple targets simultaneously becomes a key factor. While this is complex in gene-editing approaches, t is relatively simple in a non-gene-edited approach. Multiple groups have combined either miRNA- or siRNA-like sequences in an effort to inhibit multiple target-sequences together either through a natural scaffold or a synthetic one [157,158,159,160]. We have recently developed a microRNA (miRNA)-based multiplex shRNA platform, obtained by combining highly efficient miRNA scaffolds into a chimeric cluster [161]. We were able to deliver up to four shRNA-like sequences (in a plug-and-play manner) in a single vector containing the CAR and four different shRNA-like sequences targeting CD3ζ (GvHD), B2M (HLA-I/HvG) and additional combinations of either CIITA (HLA-II/HvG), CD95 (Fas receptor/inhibit apoptosis), LAG-3 (Immune-checkpoint inhibitor) and/or CD28 (co-stimulation, reduction/persistence). Interestingly, we discovered that the modulation of genes rather than gene KO is essential for certain targets (such as B2M, where a clear balance exists between removal of the HLA-I and recognition by NK-cells and the minimal expression needed to avoid NK-cell lysis and/or T-cell-mediated activation), making the method a good and easy-to-use tool for certain targets.

### 4.2. Clinical Experience with ‘Off-the-Shelf’ Allogeneic CAR-Ts

#### 4.2.1. Successes to Date

Several off-the-shelf allogeneic CAR-T products are currently under clinical evaluation in Phase I or Phase I/II studies by several groups (Table 2).

Most experience to date has been obtained with an allogeneic universal anti-CD19 CAR-T product (UCART19/ALLO-501), which is genome-edited with TALEN technology to simultaneously disrupt TRAC and CD52 genes. While TRAC is targeted to prevent GvHD risk, CD52 gene knockout aims to protect allogeneic CAR-Ts from rejection by alemtuzumab/ALLO-647, an anti-CD52 antibody used as an additional lymphodepleting agent [121]. UCART19/ALLO-501 was evaluated in two completed Phase I studies in pediatric (PALL study, NCT02808442) and adult (CALM study, NCT02746952) populations with relapsed or refractory (r/r) B-cell acute lymphoblastic leukemia (B-ALL) [162,163], initiated following successful therapy in two infants with r/r B-ALL who had relapsed after a first allo-SCT [187]. UCART19/ALLO-501 is still under evaluation in adult patients with r/r large B-cell lymphoma (LBCL) or follicular lymphoma (FL) in the ALPHA study (NCT03939026). Globally, UCART19/ALLO-501 induced antileukemic activity with an overall response rate (ORR) of 48% in these heavily pretreated populations and exhibited a manageable safety profile with moderate cytokine release syndrome (CRS) events and minimal—but still some (8% of patients)—grade 1 acute cutaneous GvHD. Two deaths in the CALM trial were considered to be associated with UCART19, and both were reported as dose-limiting toxicity [162]. Of note, between 3% and 6% of UCART19/ALLO-501 cells had translocation-associated karyotype abnormalities, without suggestion of adverse effects [162]. An evolution of UCART19/ALLO-501 was also developed and referred to as ALLO-501A, in which the safety switch rituximab recognition was removed; it is currently being evaluated in the ALPHA2 (NCT04416984) and EXPAND (NCT05714345) studies. Data from the optimal lymphodepletion regimen confirmed a good anti-tumor efficacy, with an ORR of 67% across both ALPHA and ALPHA2 studies and no GvHD reported [188].

The same TALEN technology is used in the universal anti-CD123 (UCART123) and anti-CD22 (UCART22) CAR-T product candidates evaluated in two Phase I studies involving adult patients with relapsed or refractory B-ALL (NCT04150497, BALLI-01 study) [168] or relapsed/refractory acute myeloid leukemia (NCT03190278, AMELI-01 study) [114], respectively, as well as in the anti-BCMA CAR-T product ALLO-715 currently under evaluation in the Phase 1 UNIVERSAL study involving refractory/relapsed adult multiple myeloma patients [166]. Under the optimal lymphodepleting regimen, 70.8% of patients had an objective response. The median duration of response was 8.3 months, and no cases of GvHD were reported.

The ARCUS genome-editing technology is used in the anti-CD19 allogeneic CAR-T product (PBCAR0191, Azercabtagene zapreleucel or Azer-Cel) currently under evaluation in a Phase I/II study involving relapsed/refractory non-Hodgkin’s lymphoma (NHL) and B-ALL patients (NCT03666000) and has shown promising results. Azer-Cel achieved an 83% ORR and a 61% complete response (CR) rate with a 55% durable response among evaluable patients who had relapsed following autologous CAR-T therapy (n = 18) and a 58% ORR overall (n = 61) and no GvHD reported [171]. The PBCAR19B product, an anti-CD19-targeting allogeneic CAR-T designed to evade immune rejection by host T-cell and NK-cells, was evaluated in a Phase I study (NCT04649112) and achieved a 71% ORR and a 43% CR rate [171].

CRISPR/Cas9 is also used in the CD19-targeting CTX110 product evaluated in the phase 1 CARBON trial (NCT04035434) in patients with r/r NHL. A 67% ORR was observed at the highest dose level [175]. No cases of GvHD were reported despite a high HLA mismatch between donors and patients [175]. The only case of immune effector cell-associated neurotoxicity syndrome (ICANS) of Grade 3 or higher was in a patient with concurrent HHV-6 (Human Herpes Virus). Administration of a second CTX110 infusion was well tolerated and demonstrated evidence of further clinical benefit. CTX-130, an anti-CD70 allogeneic CAR-T, was evaluated in the COBALT-LYM study (NCT05722418) in patients with T-cell lymphoma and in the COBALT-RCC study (NCT04502446) in patients with advanced clear cell renal cell carcinoma (RCC). At the highest dose level, the ORR was 71% in patients with T-cell lymphoma [178], and there was no report of GvHD in any of the 17 evaluated patients. Results in the RCC population showed an ORR of 8% (n = 13), with one patient experiencing a durable complete response maintained at 18+ months and no GvHD reported [177]. The anti-CD19 CAR-T product CB-010, engineered via CRISPR/Cas9 to knockout both TRAC and PD-1 to reduce T-cell exhaustion, is evaluated in the ANTLER Phase 1 study (NCT04637763) in patients with B-NHL and showed 94% ORR across all dose levels, which rivals autologous products [174], and no GvHD was observed (n = 16).

CAR T-cell products using non-gene-editing technologies were also evaluated in clinic. CYAD-101 is an allogeneic CAR-T candidate engineered to co-express a CAR based on NKG2D, a receptor recognizing eight different stress ligands, and an inhibitory peptide interfering with the signaling of the endogenous TCR complex. CYAD-101 was evaluated in the alloSHRINK phase I study in patients with unresectable metastatic colorectal cancer (NCT03692429). Twenty-five patients received three infusions of CYAD-101 after standard preconditioning chemotherapy (FOLFOX or FOLFIRI). No dose-limiting toxicity or GvHD were reported, nor was any patient discontinuation due to treatment-related adverse events or treatment-related adverse events greater than Grade 3. The results also showed that two patients achieved a partial response (13% ORR), including one patient with a KRAS mutation, and nine patients (60%) reached a stable disease [154,184]. In CYAD-211, a miRNA-based shRNA approach was used to silence the mRNA coding for the CD3ζ component of the TCR, co-expressed with an anti-BCMA CAR in the CYAD-211 product, which was evaluated in the phase I IMMUNICY-1 trial (NCT04613557) for the treatment of patients with r/r multiple myeloma. Clinical activity from 12 patients in the dose-escalation segment of the IMMUNICY-1 trial was encouraging, with 3 patients achieving partial response (PR), while 8 patients had stable disease (SD). Overall, CYAD-211 was well tolerated, with no dose-limiting toxicity (DLT), GvHD or neurotoxicity at the three dose-levels [185]. A CD19-targeting allogeneic CAR-T using intracellular retention of membrane proteins to prevent TCR expression at the surface was evaluated in a Phase I study (NCT04384393) in patients with NHL [156]. Data from the first eight patients demonstrated no evidence of GvHD reaction but presented encouraging activity (75% ORR).

Finally, the iPSC-derived CAR T-cell product candidate FT819 targeting CD19 was evaluated in a Phase I study (NCT04629729) in patients with B-cell malignancies and demonstrated a tolerable safety profile with no reported DLT or GvHD in the 12 evaluated patients [186].

#### 4.2.2. Challenges to Overcome

In the clinical setting, some allogeneic candidates have reached objective response rates similar to those observed in their autologous counterparts, and, apart from two patients (one infant and one adult) presenting with Grade I acute skin GvHD that was easily controlled [162], preliminary data from any of those studies showed no evidence of acute GvHD. Therefore, despite earlier concerns, the modifications made to prevent GvHD in allogeneic ‘off-the-shelf’ CAR-Ts seem sufficient to drastically reduce this risk.

In contrast, the engraftment of the allogeneic CAR-Ts has been stymied to some extent by host rejection, mediated by the recognition of non-self HLA molecules on the donor T-cell membrane, and is clearly the main concern of allogeneic CAR-Ts, as this limits their activity and duration of responses. For example, in the CALM study, although expansion of the CAR-Ts, similar to those observed with autologous CAR-Ts, was observed from day 8 to day 14 after infusion, a rapid decline was observed in most patients by day 28 [189], a limited duration of response. Cellular kinetics were also limited beyond day 28 with ALLO-715 [166], UCART123 [168], UCART122 [114], CYAD-101 [154] and CYAD-211 [155]. As a first solution, a deeper lymphodepletion through a more intense preconditioning regimen is generally used as an approach to improve the allogeneic CAR-T persistence. In addition, strategies to increase the dose of cells—either by using higher dose levels at first infusion, or by using multiple infusions—have been proposed but do not fully counteract the allorejection. The CALM study evaluated different lymphodepleting regimen (fludarabine 30 mg/m^2^ × 3 [days-7 to day-5] and cyclophosphamide 500 mg/m^2^ × 3 [day-4 to day-2] with or without alemtuzumab 1 mg/kg, 40 or 60 mg flat doses [day-7 to day-3]) and different cell doses (from 6 × 10^6^ to 2.4 × 10^8^ CAR-Ts per infusion). Nevertheless, although the dose of alemtuzumab, and altogether the intensity of the lymphodepleting regimen, had a positive impact on cell persistence post-infusion, it also increased the risk of infectious complications. In the UNIVERSAL study, one dose-limiting toxicity of grade 5 fungal pneumonia related to lymphodepletion was reported [166]. A second (consolidation) dose of ALLO-501/ALLO-501A was therefore proposed in the ALPHA and ALPHA2 studies around day 30 after first infusion to further maintain peripheral blood levels of CAR-Ts beyond day 28, with the aim to improve duration of responses [164,165]. A clinical hold was also reported for UCART123 in the ABC study after a fatality event [169], which led the Data Safety Monitoring Board to recommend lowering the dose of UCART123 cells and capping cyclophosphamide to a total dose of 4 g over 3 days. Similarly, one grade 5 event of multifocal pneumonia after ALLO-715 infusion was considered to be related to progressive myeloma and the conditioning regimen [166].

Alternative methods to prevent allorejection are currently being developed, and some of them have already been evaluated in clinical trials. The PBCAR19B product was engineered to knock-down β2M (beta-2 microglobulin) and express an HLA-E transgene to prevent allorejection [171]. Preliminary clinical results provide proof-of-concept that these modifications appeared to be effective in delaying the recovery of host T- and NK-cells. Similarly, CTX-130, an anti-CD70 allogeneic CAR-T modified to disrupt β2M and CD70 genes to reduce allorejection and fratricide, has been reported to elicit a durable complete response in a patient with RCC [177], which may suggest the approach is indeed improving the activity of allogeneic CAR-Ts even in solid tumors. CB-011, an anti-BCMA allogeneic CAR-T engineered with CRISPR/CAS12a to KO not only TRAC but also β2M and co-express a β2M-HLA-E fusion peptide, is currently being evaluated in the CaMMouflage Phase 1 study, and has demonstrated promising preclinical data leading to significant improvement in anti-tumor activity durability [174].

Safety risks related to the use of gene-editing technologies are still a big concern. Chromosomal abnormality was reported in a single patient who received a consolidation dose of ALLO-501A, which caused a clinical hold of several months of all studies with similar technology [165]. Investigations concluded that the chromosomal abnormality was unrelated to TALEN gene editing or manufacturing process but raised the question of safety of gene-edited cellular therapies.

Recently, the Food and Drug Administration (FDA) has determined that there is a risk of T-cell malignancies, applicable to all currently approved CAR T-cell therapies [190]. These concerns are raised with autologous CAR T-cells that persist for a long period of time. Allogeneic CAR-Ts suffer from a lack of persistence; hence, this safety concern does not currently apply. With improved technological advancements, the increase in allogeneic CAR T-cell persistence may lead to similar safety concerns.

## 5. Conclusions

Undoubtedly, more extensive research is required to prove the superior clinical efficacy of allogeneic CAR-Ts compared to their approved autologous counterparts, particularly in treating solid cancer, which has limited therapeutic options.

However, current evidence suggests that allogeneic CAR-Ts can efficiently overcome major hindrances that restrict access to CAR-T therapy to a wider patient population. This feasible approach stems from the emergence of suitable techniques that interrupt the endogenous TCR and mitigate GvHD, which is the primary risk for toxicity in allogeneic T- cell treatment. Genetic ablation of TCRα through targeted gene-editing techniques has become popular in this field. However, there are potential drawbacks related to double-strand DNA breaks and the manufacturing complexities which may impact cell fitness and/or yield. A noteworthy substitute exists in the form of non-gene-editing technologies, which warrant further exploration because they offer potentially safer and more adaptable choices for manufacturing next-generation CAR-Ts. Nevertheless, although those approaches seem promising to prevent the risk of GvHD, rejection of the cells following post-infusion, via HvG reaction, is the greatest challenge as of today.

Hence, the ideal ‘off-the-shelf’ allogenic CAR-T not only needs to prevent GvHD but also HvG via diverse modifications like the downregulation or disruption of genes involved in allorejection, such as B2M, CIITA or CD52. It is therefore imperative to further invest in developing technologies that allow safe administration of allogeneic CAR-Ts while improving their persistence and their efficacy and maintaining a favorable safety profile.

## Figures and Tables

**Figure 1 cells-13-00146-f001:**
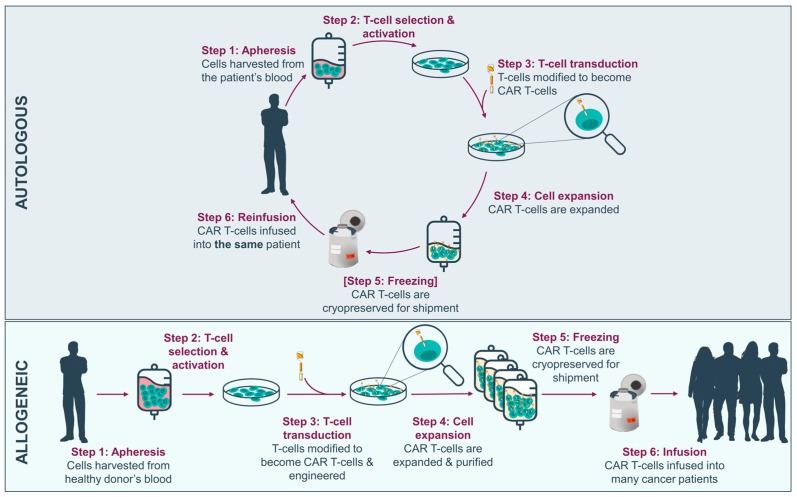
Overview of autologous versus allogeneic CAR-T manufacturing process from peripheral blood mononuclear cells.

**Table 1 cells-13-00146-t001:** Gene-editing technologies used to engineer allogeneic CAR-Ts.

	ZFN	TALEN	CRISPR/Cas9	CRISPR/Cas12a	Base-Editing
Recognition site	Zinc finger protein	RVD tandem repeat region TALE protein	Two RNA molecules (guideRNA and tracrRNA)	Single-stranded guide RNA	CRISPR/Cas dependent (Cas sequence + base-editor mRNA)
Modification pattern	Fok1 nuclease	Fok1 nuclease	Cas9 nuclease	Cas12a nuclease	Four possible transition mutations: C→T A→G T→C G→A
Target sequence size	9–18 bp	14–20 bp	20 bp- guide + PAM sequence	20 bp- guide + PAM sequence	CRISPR/Cas dependent
Specificity	Small number of positional mismatches	Small number of positional mismatches	Positional/multiple consecutive mismatches	Positional/multiple consecutive mismatches	CRISPR/Cas dependent
Targeting limitations	Difficult to target non-G-rich sites	5′ targeted base must be a T for each TALEN monomer	Recognizes 3′ G-rich Must precede a PAM sequence of 3–5 nt	Recognizes 5′ T-rich Must precede a PAM sequence of 3–4 nt	CRISPR/Cas dependent
Engineering	Requires substantial protein engineering	Requires complex molecular cloning methods	Uses standard cloning procedures	Uses standard cloning procedures	Uses standard cloning procedures
Delivery	Easy due to small size	Difficult due to large size	Moderate to difficult due to large size of SpCas9	Moderate to difficult due to large size of FnCas12a	Difficult due to large site and added complexity

PAM: protospacer-adjacent motif.

**Table 2 cells-13-00146-t002:** Engineered allogeneic ‘off-the-shelf’ CAR-Ts with published clinical experience.

Allogeneic Engineering Technology	Target Antigen	Strategy for GvHD	Strategy for HvG	Product Name	Developers	Trial Names, Phase and Number
αβ T-cells (from PBMCs)						
TALEN	CD19	Disruption of TRAC	Disruption of CD52 and use of anti-CD52	ALLO-501/UCART19	Cellectis (Paris, France); Allogene Therapeutics (San Francisco, CA, USA)	CALM Phase 1 [162,163] NCT02746952 PALL Phase 1 [162] NCT02808442 ALPHA Phase 1 [164] NCT03939026
	CD19	Disruption of TRAC	Disruption of CD52 and use of anti-CD52	ALLO-501A	Cellectis (Paris, France); Allogene Therapeutics (San Francisco, CA, USA)	ALPHA 2 Phase 1/2 [165] NCT04416984
	BCMA	Disruption of TRAC	Disruption of CD52 and use of anti-CD52	ALLO-715	Allogene Therapeutics (San Francisco, CA, USA); Cellectis (Paris, France)	UNIVERSAL Phase 1 [166] NCT04093596
	CD70	Disruption of TRAC	Disruption of CD52 and use of anti-CD52 CD70 CAR designed to avoid fratricide	ALLO-316	Allogene Therapeutics (San Francisco, CA, USA); Cellectis (Paris, France)	TRAVERSE Phase 1 [167] NCT04696731
	CD123	Disruption of TRAC	Disruption of CD52 and use of anti-CD52	UCART123	Cellectis (Paris, France)	AMELI-01 Phase 1 [114] NCT03190278 Phase 1 NCT04106076 ABC123 Phase 1 NCT03203369
	CD22	Disruption of TRAC	Disruption of CD52 and use of anti-CD52	UCART22	Cellectis (Paris, France)	BALLI-01 Phase 1 [168] NCT04150497
	SLAMF7	Disruption of TRAC	Disruption of CS1 gene to avoid fratricide	UCARTCS1	Cellectis (Paris, France)	MELANI-01 Phase 1 [169,170] NCT04142619
ARCUS	CD19	Disruption of TCR	-	PBCAR0191/Azercabtagene zapreleucel	Precision BioSciences (Durham, NC, USA)	Phase 1/2 [171] NCT03666000
	CD19	Disruption of TCR	shRNA against β2M and HLA-E transgene	PBCAR19B	Precision BioSciences (Durham, NC, USA)	Phase 1 [171] NCT04649112
	BCMA	Disruption of TCR	-	PBCAR269A	Precision BioSciences (Durham, NC, USA)	Phase 1 [172] NCT04171843
	CD20	Disruption of TCR	-	PBCAR20A	Precision BioSciences (Durham, NC, USA)	Phase 1/2 [173] NCT04030195
CRISPR/Cas9	CD19	Disruption of TRAC	-	CB-010	Caribou Biosciences (Berkeley, CA, USA)	ANTLER Phase 1 [174] NCT04637763
	CD19	Disruption of TRAC	Disruption of β2M	CTX110	CRISPR Therapeutics (Zug, Switzerland)	CARBON Phase 1/2 [175] NCT04035434
	BCMA	Disruption of TRAC	Disruption of β2M	CTX120	CRISPR Therapeutics (Zug, Switzerland)	Phase 1 [176] NCT04244656
	CD70	Disruption of TRAC	Disruption of β2M + CD70 disruption to avoid fratricide	CTX130	CRISPR Therapeutics (Zug, Switzerland)	COBALT-RCC Phase 1 [177] NCT04438083 COBALT-LYM Phase 1 [178] NCT04502446
	CD19	Disruption of TRAC	Disruption of CD52 and use of anti-CD52	CTA101	Nanjing Bioheng Biotech (Nanjing, China)	Phase 1 [179] NCT04154709 NCT04227015
	CD19/CD7	Disruption of TRAC	CD7 disruption to avoid fratricide	GC502	Gracell Biotechnologies (Suzhou, China)	Early Phase 1 [180] NCT05105867
	CD7	Disruption of TRAC	CD7 disruption to avoid fratricide	WU CART 007	Wugen (St Louis, MO, USA)	Phase 1/2 [181] NCT04984356
Cas-CLOVER™	BCMA	Disruption of TCR beta chain 1	Disruption of β2M	P-BCMA-ALLO1	Poseida Therapeutics (San Diego, CA, USA)	Phase 1 [182] NCT04960579
	FKBP12; MUC1-C	Disruption of TCR	Disruption of β2M	P-MUC1C-ALLO1	Poseida Therapeutics (San Diego, CA, USA)	Phase 1 [183] NCT05239143
Base-pair editing	CD7	Disruption of TRAC	Disruption of CD52 and CD7 to avoid fratricide	BE-CAR7	Great Ormond Street Hospital (London, UK)	Phase 1 [147] ISRCTN15323014
Peptide-based (TIM8)	NKG2DL	Negative competition with CD3ζ	-	CYAD-101	Celyad Oncology (Mont-Saint-Guibert, Belgium)	alloSHRINK Phase 1 [154,184] NCT03692429 CYAD-101-002 Phase 1 NCT04991948
miRNA-based shRNA	BCMA	Knock-down of CD3ζ	-	CYAD-211	Celyad Oncology (Mont-Saint-Guibert, Belgium)	IMMUNICY-1 Phase 1 [185] NCT04613557
Non-gene editing	CD19	Intracellular retention of TCR/CD3 complex via KDEL-tagged anti-CD3 scFv	Decreasing surface HLA-A and HLA-B by HCMV US11 protein	ThisCART19 cells	Fundamenta Therapeutics (Suzhou, China)	Phase 1 [156] NCT04384393
cytolytic T-lymphocytes (from PBMCs)						
Zinc Finger Nuclease	IL13-zetakine	Disruption of the glucocorticoid receptor	Use of dexamethasone	GRm13Z40-2	City of Hope (Duarte, CA, USA)	Phase 1 [111] NCT01082926
αβ T-cells (from iPSCs)						
CRISPR/Cas	CD19	Disruption of TRAC	-	FT819	Fate Therapeutics (San Diego, CA, USA)	Phase 1 [186] NCT04629729

iPSC: Induced pluripotent stem cells; PBMC: peripheral blood mononuclear cells.

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
