# Peer review of "Allogeneic CAR-T Therapy Technologies: Has the Promise Been Met?"

_cells, 2024, doi:10.3390/cells13020146_

Round 1

Reviewer 1 Report

Comments and Suggestions for Authors

Nicely written review. Overall, I enjoyed reading it. 

The authors made a literature review on allogeneic CAR-T cells as an off-the-shelf source for cell therapies. The review is well-written, original, and of interest to the field. My only suggestion to improve the manuscript would be to:  

1. discuss the recent concern on malignancies and put into perspective of the allogeniec CAR T approach 
https://www.fda.gov/vaccines-blood-biologics/safety-availability-biologics/fda-investigating-serious-risk-t-cell-malignancy-following-bcma-directed-or-cd19-directed-autologous#:~:text=FDA%20has%20determined%20that%20the,several%20products%20in%20the%20class.

2. Describe the risk associated with DSB, i.e  chromosomal truncation and chromosomal loss (https://doi.org/10.1016/j.cell.2023.08.041)

3. Mention the risk of promiscuous deamination effects with BE in both DNA and RNA

Reviewer 2 Report

Comments and Suggestions for Authors

The paper is potentially interesting for publication in a journal and contains appropriate diagrams and tables. This is a review paper. However, some changes are necessary in order to better explain the phenomena.

At the same time, it is necessary to add appropriate citations in certain places to confirm the facts presented in the paper.

1. In the introductory part about the importance of immunotherapy, a paper indicating the importance of immunotherapy should be added immediately after the introductory sentence line 25: PMID: 37296617

2. in the section entitled: 2. Source of allogeneic cells, original works should also be added that show the origin of cells from the periphery as well as the advantage in relation to classical transplantation that uses various processes of obtaining CD34 cells from the bone marrow: doi.org/10.1016/j .tranon.2023.101811

3. when describing cytokine-induced killer (CIK) cells, the full name should be written, as well as their characterization during their appearance - line 235

4. In this connection, add a reference showing the possibility that cytokines stimulate NK cells isolated from patients with tumors in addition to references 60, 61, PMID: 33270021
